# Self-Aware Reinforcement Learning for Improving LLMs with Minimal Data

## Abstract

Reinforcement learning (RL) has demonstrated potential in enhancing the reasoning capabilities of large language models (LLMs), but such training typically demands substantial efforts in creating and annotating data. In this work, we explore improving LLMs through RL with minimal data. Our approach alternates between the LLM proposing a task and then attempting to solve it. To minimize data dependency, we introduce two novel mechanisms grounded in *self-awareness*: (1) self-aware *difficulty prediction*, where the model learns to assess task difficulty relative to its own abilities and prioritize challenging yet solvable tasks, and (2) self-aware *limit breaking*, where the model recognizes when a task is beyond its capability boundary and proactively requests external data to break through that limit. Extensive experiments on nine benchmarks showing a 53.8% relative improvement with less than 1.2% extra data demonstrate the efficacy of self-aware RL and underscore the promise of self-evolving agent training. Our code is available at https://anonymous.4open.science/r/SARL-B7FE/.

## 1 Introduction

Reinforcement Learning (RL) has emerged as a crucial approach for enhancing the reasoning abilities of large language models (LLMs), especially in tasks like mathematical problem solving and code generation, where the correctness of generated outputs can be rigorously verified (Lambert et al., 2024; Guo et al., 2025; Jaech et al., 2024; Team et al., 2025). However, these improvements often come at the cost of requiring vast amounts of high-quality data. The data curation processes, which heavily rely on human experts to create tasks and annotate solutions, are prohibitively costly and time-consuming, creating a significant bottleneck in LLM advancements.

One promising solution is to empower LLMs to generate their own tasks, enabling self-improvement through RL (Zhao et al., 2025; Qi et al., 2025; Yang et al., 2025; Wang et al., 2025; Fang et al., 2025). By actively participating in their own learning process—generating tasks and attempting to solve them—LLMs can establish a self-improving loop of data creation and model refinement, significantly reducing the need for manually curated data. While initial efforts have shown promise, this line of research faces two critical challenges:

First, *the model may fail to generate appropriately challenging tasks*. If the tasks are too easy, the model fails to engage in deeper reasoning and exploration. Conversely, if the tasks are too difficult, the model receives little useful feedback about what went wrong or how to improve. This issue is further complicated by the dynamic nature of the self-improvement loop, as the model's evolving capabilities require that task difficulty be continuously adjusted. Second, *the self-improving loop is prone to stagnation, failing to continuously proposing challenging tasks*. While the model can improve through RL, the self-improving loop is fundamentally constrained by the base model's inherent capabilities. As the loop progresses, the difficulty of the generated tasks may plateau, once the model has exhausted its current knowledge. Consequently, the model may encounter a point of stagnation, unable to advance further or surpass its inherent capability limits.

This paper argues that the key to addressing these challenges lies in *self-awareness*—the ability to understand one's own strengths, weaknesses, and evolving capabilities. A task that is difficult for the base model may later become trivial as its abilities improve. Therefore, the model must be acutely aware of its current abilities to accurately assess the difficulty of its generated tasks in relation to its

state. Furthermore, overcoming the model's capability limits and escaping stagnation also require a clear understanding of its own capability boundaries.

To this end, this paper introduces a novel paradigm for improving LLMs with minimal data: self-aware RL. This paradigm features two key mechanisms. First, through self-aware *difficulty prediction*, the model not only generates a task but also predicts the difficulty of the task considering its current state. To achieve accurate predictions, the model learns to align its difficulty estimates with its actual success rate. This enables the creation of an adaptive curriculum that prioritizes problems with an appropriate level of difficulty, tailored to the current capabilities of the LLM. Second, through self-aware *limit breaking*, the model *occasionally* proactively seeks external guidance, such as requesting a correct solution, when it identifies a generated task to be highly valuable for improvement but hardly solvable given its current capability. This is achieved by having the model assess the novelty (via perplexity) and difficulty (via its own prediction) of a generated task. This allows the LLM to surpass its inherent capability limits while minimizing reliance on external data, resorting to it only when necessary.

For evaluation, we train the LLMs with self-aware RL equipped with python code interpreter that provides verifiable outcome signals, and then evaluate the LLMs across a range of reasoning tasks the models have not encountered during our training. In our experiments, we implement self-aware RL based on Qwen2.5-Coder-3B, and consider nine benchmarks across mathematical reasoning and code generation in the evaluation. Our results demonstrate that self-aware RL can significantly improve the pre-trained model's performance on both out-of-distribution tasks (code generation benchmarks) and out-of-domain tasks (mathematical reasoning benchmarks), achieving a relative performance gain of **53.8%** on average on mathematical reasoning benchmarks. Specifically, self-aware RL got significant improvements on a lot of widely used benchmarks: **29.8%** on MATH500, **77.8%** on AMC'23, **82.4%** on OlympiadBench, and **22.3%** on LiveCodeBench. We further analyze the training behavior, and conduct fine-grained ablation studies to learn the impact of different components.

To summarize, our key contributions are as follows:

- We establish connections between self-awareness and the self-improvement of LLMs, highlighting that self-awareness is a crucial mechanism for efficiently enhancing LLMs, while reducing reliance on external data.

- We present a novel RL paradigm for improving LLMs with minimal data, enabling the model to (i) generate appropriately difficult tasks that align with its current abilities, and (ii) proactively seek external guidance to surpass its inherent capability limits when necessary.

- We conduct extensive experiments across nine benchmarks in mathematical reasoning and code generation, showing that our approach substantially boosts the base model's performance and generalization abilities, thereby validating the effectiveness of the self-aware RL paradigm.

## 2 RELATED WORKS

**Reinforcement Learning with Verifiable Reward (RLVR).** RLVR is an emerging paradigm (Guo et al., 2025; Lambert et al., 2024; Jaech et al., 2024; Team et al., 2025) in LLM alignment. Tulu3 (Lambert et al., 2024) was the first LLM tuned with RLVR to improve math and instruction following capabilities. Deepseek-R1 (Shao et al., 2024; Guo et al., 2025) was trained with Group Relative Policy Optimization (GRPO), a novel RLVR algorithm which uses inner group relative reward to replace the value model in traditional RL algorithms, boosting the training efficiency. Many following works tried to stabilize GRPO training. DAPO (Yu et al., 2025) used a higher clipping threshold and proposed dynamic sampling to better utilize the positive reward signal. GSPO (Zheng et al., 2025) updated the token-level importance sampling in GRPO to sequence-level, mitigating the bias introduced by imbalanced response lengths. Dr.GRPO Liu et al. (2025) identified that GRPO tends to overly penalize shorter, incorrect responses, which misguides the LLM to produce longer yet incorrect responses. Similar to GSPO, Dr.GRPO fixed this issue by considering a sequence-level reward instead of a token-level one. Due to the significant performance gain brought by RLVR, it is widely adopted in most advanced Large Reasoning Models (LRM) (Team et al., 2025). However, while RLVR does not require human crafted responses for

imitation learning like in Supervised Fine Tuning (SFT), a large set of high-quality tasks are still necessary to incentivize diverse and effective reasoning patterns.

**Self-evolving RL.** To overcome the dependency on vast data, self-evolving RL is proposed to create agents that autonomously enhance their capabilities with minimal human oversight. Wang et al. (2025) proposed to build a unit tester co-evolving with the code generation agent to improve the overall generation quality. WebEvolver (Fang et al., 2025) introduced a dynamic world model simulating web navigation to help exploration in the web environment. Absolute Zero Reasoner (Zhao et al., 2025) proposed a generator agent automatically generating coding tasks to train the policy model without any input data. Similarly, self-challenging agent Zhou et al. (2025) designed a geneator agent that can interact with the environment to simulate a user query as the task fed into a customer agent. WebRL (Qi et al., 2025) proposed to generate new tasks from unsuccessful attempts of a web agent, expanding the data coverage to improve the decision-making capability of the agent. ZeroGUI (Yang et al., 2025) adopted VLM-based automatic task generation to produce diverse training goals. However, these approaches generate tasks without awareness of the agent's own capabilities, leading to inefficient training curricula. The generated tasks are often misaligned with the agent's learning frontier, being either too trivial or difficult.

**Curriculum RL in LLM Reasoning.** Curriculum learning, which involves structuring training data from easy to hard, has emerged as a powerful technique for cultivating advanced reasoning in LLMs. Typical curriculum learning methods focus on dynamically selecting or filtering problems from a larger pool to maintain an optimal level of difficulty. Recently, curriculum learning has also caught much attention from LLM community due to the reliance on proper utilization of available data in LLM training. Balanced online difficulty filtering (Bae et al., 2025) proposed a novel mechanism, balanced filtering, that focuses on tasks with medium level of hardness to maximize the effectiveness of GRPO. Adaptive Difficulty Curriculum Learning (Zhang et al., 2025) imitates human learning strategy and periodically re-estimates the difficulty within upcoming data batches to keep aligned with model's capabilities. Self-Evolving Curriculum (Chen et al., 2025) learns an additional curriculum policy concurrently with the RL fine-tuning process to maximize the learning progress. Curriculum Reinforcement Fine-tuning (Deng et al., 2025) proposed a difficulty-aware reward design ensuring steady progression of model capabilities. By guiding models through a structured learning path, these methods effectively foster the development of complex, multi-step reasoning capabilities. Even though, these curriculum learning approaches directly improve typical RL methods. The conjunction of curriculum learning and self-evolving RL is largely underexplored. In this paper, we explore a solution combining the advantages of both, showing that the curriculum design is especially important in self-evolving RL training.

## 3 PRELIMINARIES

In this paper, we denote the LLM parameterized by $\theta$ as a policy model $\pi_\theta$, which is also used to refer to an LLM agent driven by $\pi_\theta$ for simplicity. We use $x$ to denote the task that a solver agent solves with response $y$, where $x$ and $y$ are both sequences of tokens with $t$-th token in $x$ denoted as $x_t$, and the length of $x$ denoted as $|x|$.

REINFORCE++ (Hu et al., 2025)is an advanced RL algorithm for LLM post-training. Its learning objective on a sampled pair $(x, y)$ is given by:

$$\mathcal{J}(\theta|x, y) = \frac{1}{|y|} \sum_{t=1}^{|y|} \min\left(w_t(\pi_\theta)\hat{A}_t, \text{clip}\left(w_t(\pi_\theta), 1 - \varepsilon, 1 + \varepsilon\right)\hat{A}_t\right) \tag{1}$$

where $w_t(\theta)$ is the importance sampling ratio, and $\hat{A}_t$ denotes the normalized advantage term:

$$\hat{A}_t = \frac{r - \text{mean}_{\text{batch}}(r)}{\text{std}_{\text{batch}}(r)} \tag{2}$$

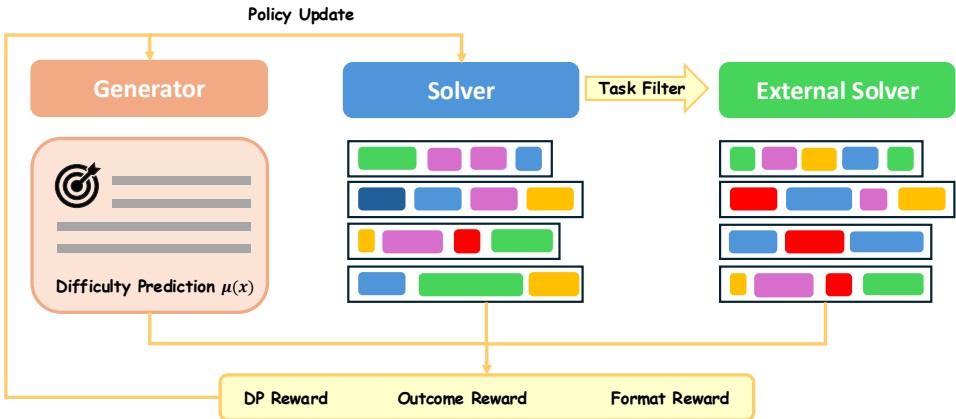

Figure 1: The overview of self-aware RL. The generator agent first generates the task, with a predicted success rate $\mu(x)$ (Section 4.1). The solver agent will then generates reasoning paths. If none of these generated reasoning paths are correct, which means no update are available, the task will be filtered by a task filter to determine whether it is of enough value to be processed by an external solver (Section 4.2). Finally, the collected difficulty prediction reward, outcome reward, and format reward will be aggregated to calculate the policy update (Section 4.3).

# 4 METHOD

The general framework of self-aware RL can be viewed in Figure 1. We design self-aware RL as a self-evolving training loop, where a generator agent and a solver agent iteratively generate and solve new tasks to improve themselves. Due to the stability and accessibility of coding environment, we implement self-aware RL on code generation tasks, while any other environment that can check the correctness of the agent's response also applies, *e.g.*, the verifiable rewards gym adopted by Kimi-K2 (Team et al., 2025). We use the python code interpreter to produce verifiable reward signals. Given a code snippet, the solver agent is asked to predict the outcome after execution, and a matching outcome will receive a positive outcome reward. Therefore, the generator agent can generate a new code snippet with accessible execution outcome. With the generated tasks, self-aware RL improves the solver agent with typical RLVR methods.

Specifically, self-aware RL incorporates *difficulty prediction* and *limit breaking* to improve the self-awareness of the agent. LLM agents based on pre-trained LLMs are not inherently self-aware, which hinders efficient self-evolution. With a better understanding of their own capability boundaries, agents can generate properly challenging tasks, which has been validated to be beneficial to RL training (Bae et al., 2025; Chen et al., 2025). Meanwhile, fully on policy RL training can be extremely challenging in some cases where the agent is strictly limited by its inherent capability ceiling. External guidance is necessary here to break the limitation. However, it is not efficient to seek external guidance blindly on every unsolvable tasks, which degrades to naive supervised fine-tuning (SFT) requiring heavy human inspections. Self-awareness can assist in identifying highly valuable tasks where external guidance is necessary and effective, mitigating the need for vast, well-grounded supervision.

- Self-aware *difficulty prediction* guides the generator agent to learn to predict the difficulty of the generated tasks.
- Self-aware *limit breaking* identifies cases where the solver are limited by its inherent reasoning capability, and breaks it with external guidance.

## 4.1 SELF-AWARE DIFFICULTY PREDICTION

Problems at a proper level of difficulty can maximize the learning progress (Chen et al., 2025; Bae et al., 2025). However, guiding the generator to produce suitable tasks requires a good understanding of the LLM's own capability which entails non-trivial challenges. LLMs are rarely trained to improve the capability of understanding their own boundary as it is not feasible to do so in large scale

pre-training with preset human inspection. Consequently, while being a great task solver, LLMs are not naturally good at understanding themselves. Our intuition is empirically confirmed through evaluating pre-trained LLMs' performance in predicting task solving success rates. As discussed in Figure 3, we found that the base model is initially bad at predicting the task solving success rates, *i.e.*, LLMs are not inherently aware of its own capability boundary.

Therefore, we propose self-aware *difficulty prediction* to enhance the generator agent's capability of understanding its own knowledge boundaries and generate proper tasks benefiting the self-evolving loop. First, we enable self-aware self-aware RL by asking the generator agent to explicitly reason about the complexity of the generated task and predict how many trials would be correct among $N$ rollouts generated by the solver agent. For instance, assume that the generator agent predicts that for a generated task (we denote the task as $x$) to be solved, the solver agent can generate 5 correct answers out of $N = 8$ rollouts, we will have the predicted success rate $\mu(x) = \frac{5}{8}$. The predicted success rate $\mu(x)$ should be aligned with the ground truth success rate $\hat{\mu}(x)$. When rollouts are sampled from the solver agent, *difficulty prediction* collects the sampled rollouts and calculate the ground truth success rate, which is further utilized in the reward signal for the generator. Specifically, given the actual success rate $\hat{\mu}(x)$, and the predicted success rate $\mu(x)$, the reward function is:

$$R_{\text{dp}}(x) = 1 - |\hat{\mu}(x) - \mu(x)| \tag{3}$$

When the generator predicts a close enough success rate, it will receive a high reward from it.

### 4.2 SELF-AWARE LIMIT BREAKING

Even with proper tasks, agents are still limited by their own capability. When the solver fails to solve a task with multiple trials, it will not get any improvement since no positive guidance is acquired. Previous practices to deal with this challenge have focused on injecting new knowledge and abilities into the LLM during training, a process that commonly involves human inspection. As LLM training is extremely costly, intervention from human experts will not be frequent, which results in a widely adopted training paradigm: data collection, training, evaluation, and repeat. This less dynamic and timely intervention makes the benefit from external guidance take effect much later than the time point when the LLM actually requires the guidance.

To mitigate this issue, self-aware RL adopts self-aware *limit breaking* to offer effective guidance immediately when the solver agent fails to solve a valuable task. As shown in Figure 1, for any task, when all reasoning paths sampled from the solver agent are incorrect, *limit breaking* will first check the task utility representing whether the task is significant in improving the agent. Once validated, external guidance will be obtained by querying a stronger external solver. Correct guidance will be adopted to improve the solver agent, breaking the inherent ability limitation.

We first introduce how to identify valuable tasks. For clarity, we use an indicator variable $\mathbb{I}_x$ to denote whether a task is selected as valuable enough to acquire external guidance. The selection is a random event that occurs with a probability $p(x)$ depending on the task itself. Formally, $\mathbb{I}_x$ is a Bernoulli random variable defined as

$$\mathbb{I}_x = \begin{cases} 1, & \text{with probability } p(x) \\ 0, & \text{with probability } 1 - p(x) \end{cases}$$

To obtain suitable $p(x)$, we design a task utility mechanism that assigns different levels of utility score to tasks reflecting their significance. For task utility, we consider two dimensions: the difficulty and the novelty. For difficulty, we utilize the predicted success rate from *difficulty prediction* as this score reflects the general review of complexity from the generator agent. We inversely use $1 - \mu(x)$ to measure the difficulty level of task $x$. To measure the novelty of a task, we utilize the token-level perplexity from the solver agent. The utility score $\omega$ of task $x$ can be formulated as follows:

$$\omega_{\text{difficulty}}(x) = 1 - \mu(x)$$

$$\omega_{\text{novelty}}(x) = -\frac{1}{|y|} \sum_{i=1}^{|y|} log \pi_{\theta_{\text{old}}}(y_i|x, y_{<i}) \tag{4}$$

A higher novelty score indicates that the current policy is more confused about the task description, which can be viewed as a signal the solver agent is not familiar with the task. While we only consider two kinds of utility measurement here, it is possible to take more measurement into consideration when transferred to other domains.

In order to minimize the amount of acquired external guidance used to save cost, we only acquire external guidance on filtered-out high-utility tasks. Given that the tasks are dynamically generated and the agent's capability may differ between runs, it is natural to calculate the utility relatively by comparing a task to other tasks in that run. To achieve this, we maintain a task buffer $\mathcal{B}$ that records recent tasks seen by the solver agent. The buffer is implemented as a FIFO queue, which only stores a fixed number of recently generated tasks.

$$\mathcal{B} = \{x_1, x_2, ..., x_{|\mathcal{B}|}\} \tag{5}$$

Tasks with top-level utility will be assigned a higher $p(x)$. To achieve this, we first obtain the z-score of the task utility among the recorded tasks, and assign a higher $p(x)$ to tasks with higher z-score. This procedure is implemented as follows:

$$\omega(x) = [\omega_{\text{difficulty}}(x), \omega_{\text{novelty}}(x)]$$

$$z(x) = \frac{1}{|\omega|} \sum_{i=1}^{|\omega|} \frac{\omega(x)_i - \mathbb{E}_{x \sim \mathcal{B}}[\omega(x)_i]}{\text{std}_{x \sim \mathcal{B}}[\omega(x)_i]} \tag{6}$$

$$p(x) = \Phi\left(\gamma\left(z(x) + \Phi^{-1}(\tau)\sqrt{1 + \frac{1}{\gamma^2}}\right)\right) \tag{7}$$

where $\omega(x)_i$ denotes the $i$-th dimension of $\omega(x)$. In this case, $\omega(x)_0 = \omega_{\text{difficulty}}(x)$, and $\omega(x)_1 = \omega_{\text{novelty}}(x)$. $\Phi$ is the cumulative distribution function (CDF) of standard normal distribution, $\gamma > 0$ is a sharpness factor and $\tau$ is the probability factor, which makes $\mathbb{E}[p(x)] = \tau$. By tuning $\tau$, we can control the number of tasks that require external guidance. A higher sharpness factor $\gamma$ will enlarge $p(x)$ for a task with high $z(x)$.

### 4.3 Self-aware RL training pipeline

Finally, we can combine all components to build the self-aware RL training pipeline. For the generator agent, we use $R_{\text{dp}}$ as the reward. Similar to typical RLVR settings, we include a format reward $R_{\text{format}} \in \{0, 1\}$ to guide the agent responses following the *difficulty prediction* template. A positive $R_{\text{format}}$ indicates that the agent's response passes the format check. We formulate the reward for the generator agent as follows:

$$R_{\text{generator}} = R_{\text{format}}R_{\text{dp}} + R_{\text{format}} \tag{8}$$

For the solver agent, we apply typical RLVR method, and adopt the binary outcome reward $R_{\text{outcome}} \in \{0, 1\}$ as the reward signal. A positive outcome reward indicates that the agent successfully solves the task. We also adopt a format reward for the solver agent, with a different dialogue template to the generator agent. We formulate the reward for the solver agent as follows:

$$R_{\text{solver}} = R_{\text{format}}R_{\text{outcome}} + R_{\text{format}} \tag{9}$$

For a training sample $(x, y)$ including one generated task from the generator agent and one sampled response from the solver agent, the learning objective is formulated in Equation 1. Therefore, the learning objective for self-aware RL is:

$$\mathcal{J}_{\text{self-aware RL}}(\theta) = \mathbb{E}\left[\mathcal{J}(\theta|x, y)\right], \ y \sim \pi_\theta^{1 - \mathbb{I}_x} \pi_{\theta_{\text{external}}}^{\mathbb{I}_x} \tag{10}$$

The training pipeline is compatible with different RL algorithms. In our implementation, we adopt REINFORCE++ (Hu et al., 2025) as the RL algorithm for verification.

# 5 EXPERIMENT

## 5.1 EXPERIMENTAL SETUP

**Datasets** We conduct our evaluation on 6 mathematic reasoning datasets and 3 coding benchmarks: MATH500 (Hendrycks et al., 2021), AIME (AIME'24 and AIME'25), Minerva (Lewkowycz et al., 2022), AMC'23, OlympiadBench (He et al., 2024), MBPP++, HumanEval++ (Liu et al., 2023), and LiveCodeBench (Jain et al., 2024).

**Training Setup** We implement self-aware RL based on verl (Sheng et al., 2024), an open-sourced reinforcement learning pipeline widely used in developing LLM RLVR frameworks. We made necessary adjustment to make it compatible with our GPU servers, with detailed hyperparameter configuration listed in Appendix A.1. We validate self-aware RL on a widely used open-sourced LLM, Qwen2.5-Coder-3B (Hui et al., 2024) given its superior coding capability which serves as a good starting point. We use Qwen2.5-Coder-32B as the external policy model. By default, we set hyperparameter in *limit breaking* as $\tau = 0.1$ and $\gamma = 5$. Due to the inaccurate difficulty prediction at the beginning, we diable *limit breaking* for the first 50 steps, which is further explained in Section 5.3.

## 5.2 EXPERIMENTAL RESULTS

Table 1: Comparing self-aware RL against baselines.

| Method | MATH500 | Minerva | AIME | Olympiad | AMC'23 | Math Avg | HumanEval++ | MBPP++ | LCB | Code Avg |
|---|---|---|---|---|---|---|---|---|---|---|
| Coder-3B | 49.0 | 17.6 | 0.0 | 15.9 | 22.5 | 21.0 | 67.7 | 66.7 | 19.3 | 51.2 |
| AZR | 43.6 | 21.0 | 1.7 | 20.4 | 25.0 | 22.3 | **71.3** | 66.9 | 20.8 | 53.0 |
| self-aware RL | **63.6** | **25.4** | **3.3** | **29.0** | **40.0** | **32.3** | 70.7 | **67.5** | **23.6** | **53.9** |

We compare self-aware RL to baselines on nine benchmarks in Table 1. Our results indicate that self-aware RL significantly outperform previous baselines. Specifically, we observe that self-aware RL outperforms Qwen2.5-Coder-3B by 53.8% on average on mathematic benchmarks, and by 5.3% on coding benchmarks. The improvement of self-aware RL on coding benchmarks is not as significant as that on mathematical benchmarks as Qwen2.5-Coder-3B is originally strong in coding tasks, leaving little space for further improvement. The obvious improvement on mathematic benchmarks can be attributed to the acquisition of stronger general reasoning capability during RL training with complex reasoning trajectories. Notably, self-aware RL only queried external guidance on 157 out of 12,800 (=1.23%) tasks.

## 5.3 ANALYSIS

**Training reward** As shown in Figure 2, self-aware RL received higher training reward in comparison to the baseline AZR, which demonstrates the performance improvement brought by self-aware RL. Note that the training reward of self-aware RL is lower at the first, which can be explained by that we adopted different dialogue template, and the policy model needs further tuning to get fitted. After 50 steps, self-aware RL received higher consistently increasing training reward.

**Difficulty prediction** We record the difficulty prediction accuracy measured by $R_{\mathrm{dp}}$ in Figure 3. Observe that the pre-trained model at the first step performs poorly on predicting its own accuracy on the generated tasks, which supports our intuition discussed in Section 4.1 that LLMs are not trained to understand their own capability boundaries. After training with self-aware *difficulty prediction* for around 50 steps, the accuracy has significantly improved from 0.2 to over 0.6. Based on this observation, in our implementation we did not enable self-aware *limit breaking* until the 50-th step, since precise measurement of task utility requires accurate prediction on the difficulty of generated tasks.

**Rollout accuracy** In Figure 4 we recorded the accuracy of rollouts sampled from the solver agent. The initially high accuracy indicates that the generator agent did not create sufficiently challenging tasks. As training progresses, the accuracy gradually decreases and stabilizes around 0.6. This observation supports our intuition from two perspectives: (i) the pre-trained base model are not

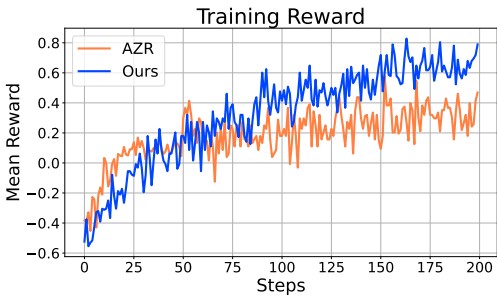
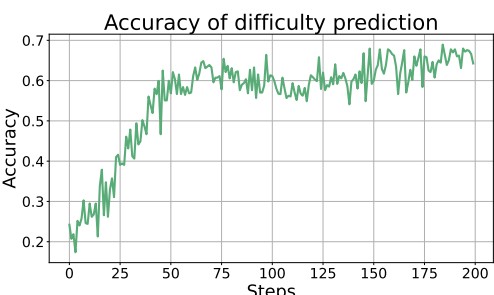

Figure 2: The training Reward of self-aware RL stably increases as the training continues. The reward is lower at the first few steps since the dialogue template is more complicated in comparison to the baseline. After the agent has been fitted to the new dialogue template, the training reward of self-aware RL quickly increases and surpasses the baseline reward.

Figure 3: Accuracy of difficulty prediction. The generator agent driven by the pre-trained base model performs poorly on the task of difficulty prediction (Section 4.1), which is shown by the low accuracy at the first step. After being tuned for 50 steps, the generator agent performs much better. Note that the accuracy shown in this figure is measured by the difficulty prediction reward in Equation 3.

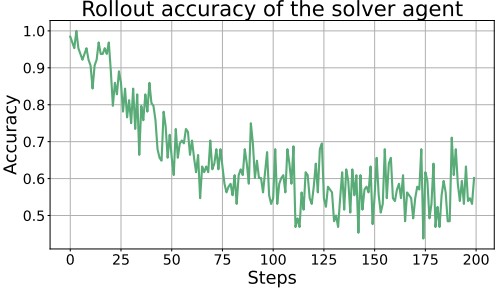
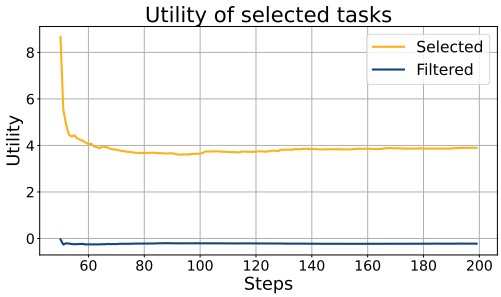

Figure 4: Accuracy of rollouts generated by the solver agent. The accuracy is initially high, reflecting that the generator did not generate challenging tasks without training. As the training continues, the difficulty of generated tasks increases and the rollout accuracy gradually decreases, was finally stabilized around 0.6.

Figure 5: Utility score $z(x)$ of selected and unselected tasks. Selected tasks should be of high utility, and will be proceeded to the external solver. While unselected tasks are of lower utility and are discarded.

good at generating appropriate tasks without specific training, and (ii) after training with self-aware RL, the generator can adaptively generate tasks at a suitable difficulty level.

**Utility of selected tasks and unselected tasks** In self-aware *limit breaking*, the solver agent will be trained on selected high utility tasks to break its capability upper bound. We compare the utility of selected and unselected tasks in Figure 5, which shows that the utility score of selected tasks are significantly higher than that of unselected tasks. This observation demonstrates that the task filter in self-aware *limit breaking* successfully distinguishes high utility tasks from low utility ones.

## 5.4 ABLATION STUDY

We conduct fine-grained ablation study to validate the effectiveness of each components in the design of self-aware RL. In the ablation study we consider three dimensions: (i) the effectiveness of *difficulty prediction* and *limit breaking*; (ii) the frequency of querying external guidance in *limit breaking*; and (iii) the effectiveness of utility ranking of *limit breaking*.

Table 2: Ablation study: comparing self-aware RL to self-aware RL w/o *limit breaking* (denoted as self-aware RL$^-$).

| Method | MATH500 | Minerva | AIME24+25 | Olympiad | AMC23 | Math Avg | HumanEval++ | MBPP++ | LCB | Code Avg |
|---|---|---|---|---|---|---|---|---|---|---|
| AZR | 43.6 | 21.0 | 1.7 | 20.4 | 25.0 | 22.3 | **71.3** | 66.9 | 20.8 | 53.0 |
| self-aware RL$^-$ | 59.6 | 23.9 | **3.3** | 25.8 | 27.5 | 28.0 | 70.1 | 66.9 | 22.9 | 53.3 |
| self-aware RL | **63.6** | **25.4** | **3.3** | **29.0** | **40.0** | **32.3** | 70.7 | **67.5** | **23.6** | **53.9** |

We first compare self-aware RL to self-aware RL w/o *limit breaking* denoted as self-aware RL$^-$ in Table 2. Observe that self-aware RL$^-$ can still outperform the AZR baseline by 25.6% on mathematical reasoning benchmarks, which is still a considerable improvement. This ablation study indicates the agent can benefit from being trained to improve the self-awareness on how likely it can successfully solve the task, which perfectly support out intuition that an LLM can generate problems more suitable for improving itself if it knows itself better.

Table 3: Ablation study: varying the amount of external guidance in *limit breaking*.

| $\tau$ | MATH500 | Minerva | AIME24+25 | Olympiad | AMC23 | Math Avg | HumanEval++ | MBPP++ | LCB | Code Avg |
|---|---|---|---|---|---|---|---|---|---|---|
| 0 | 59.6 | 23.9 | **3.3** | 25.8 | 27.5 | 28.0 | 70.1 | 66.9 | 22.9 | 53.3 |
| 0.05 | 59.6 | 24.6 | 1.7 | **30.7** | 35.0 | 30.3 | 71.3 | 66.7 | 23.2 | 53.7 |
| 0.1 | **63.6** | **25.4** | **3.3** | 29.0 | **40.0** | **32.3** | 70.7 | 67.5 | **23.6** | **53.9** |
| 0.2 | 57.2 | 23.2 | 0.0 | 23.9 | 35.0 | 27.9 | **71.3** | **67.5** | 22.1 | 53.6 |

**Ablating the amount of external guidance** In *limit breaking*, we propose to seek for external guidance on high-quality yet challenging tasks that cannot be solved directly to break through its intrinsic capability ceilings. We vary the hyperparameter $\tau$ controlling the frequency of external guidance to learn the behavior of self-aware RL with different level of external guidance. As shown in Table 3, we achieved the highest performance gain when we set $\tau = 0.1$. For a smaller $\tau$, the improvement is still obvious. However, we observe that with a larger $\tau$, the improvement from external guidance becomes ignorable. This can be explained by that overly learning from external guidance will corrupt the original reasoning pattern of the solver, which may requires further adjustment to stabilize the training process.

Table 4: Ablation study: validating the effectiveness of utility ranking.

| $\phi$ | MATH500 | Minerva | AIME24+25 | Olympiad | AMC23 | Math Avg | HumanEval++ | MBPP++ | LCB | Code Avg |
|---|---|---|---|---|---|---|---|---|---|---|
| self-aware RL$^-$ | 59.6 | 23.9 | **3.3** | 25.8 | 27.5 | 28.0 | 70.1 | 66.9 | 22.9 | 53.3 |
| self-aware RL | **63.6** | **25.4** | **3.3** | **29.0** | **40.0** | **32.3** | 70.7 | 67.5 | **23.6** | **53.9** |
| Shuffled | 59.2 | 23.9 | 1.7 | 27.1 | 30.0 | 28.4 | **72.0** | **67.7** | 21.2 | 53.6 |

**The effectiveness of utility ranking** In *limit breaking*, problems are ranked according to its utility score and high-utility problems are considered to be critical in improving the agent. We validate the effectiveness of utility ranking by randomly shuffling the sorted problems. Table 4 shows that random shuffled utility can only achieve ignorable improvement from 28.0% to 28.4% on mathematic benchmarks, and from 53.3% to 53.6% on coding benchmarks. Without proper utility ranking, *limit breaking* cannot gain obvious improvement by querying external guidance casually.

## 6 CONCLUSION

In this paper, we tackled the In this paper, we propose self-aware RL, a self-evolving framework that enables an agent to efficiently guide its own learning by leveraging self-awareness. By learning to be self-aware and recognize its own capability limits to proactively request minimal external data, self-aware RL overcomes the common failure modes of naive self-training. Our experiments validate this approach, showing that self-aware RL achieves a 53.2% relative performance gain while using less than 1% of the extra data required by conventional methods. These findings underscore the potential of self-evolving agents to reduce reliance on large-scale human annotation. This work suggests a shift from simply building larger models on large datasets, to creating smarter learners that understand their own knowledge boundaries and adapt to it. Future research could extend this self-aware learning framework to more complex, interactive domains.

## ETHICS STATEMENT

This paper does not involve human subjects, sensitive personal data, or other ethical risks. The datasets used are synthetic, and no privacy or ethical concerns are associated with this study.

## REPRODUCIBILITY STATEMENT

We have made significant efforts to ensure the reproducibility of our work. The implementation of our proposed algorithms is submitted as anonymous supplementary materials, providing complete source code for reproduction of our experiments. The process of dataset generation is described in detail in the main paper. Together, these materials ensure that the results presented in this paper can be reliably reproduced.

## STATEMENT ON THE USE OF LARGE LANGUAGE MODELS

During the preparation of this paper, we use LLMs as a writing assistant to polish up some expressions. Its role was strictly limited to improving grammar, clarity, and readability. The LLM was not used for any substantive aspect of this research, such as the generation of core ideas, the development of the methodology, code implementation, data analysis, or the formulation of conclusions.

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

# A   ADDITIONAL EXPERIMENTAL SETUP

## A.1   TRAINING SETUP

Our experiments are conducted on a server with 4 NVIDIA H-200 GPUs. We run the experiment for 200 steps. The code environment is implemented based on the QWQ python executor (Team, 2025). We list the key hyperparameter configuration of verl as follows.

Table 5: Hyperparameter Configurations

| Parameter | Value | Parameter | Value |
|---|---|---|---|
| `train_batch_size` | 64 | `max_prompt_length` | 6144 |
| `max_response_length` | 8096 | `lr` | $1 \times 10^{-6}$ |
| `ppo_mini_batch_size` | 128 | `ppo_micro_batch_size_per_gpu` | 16 |
| `ulysses_sequence_parallel_size` | 4 | `log_prob_micro_batch_size_per_gpu` | 64 |
| `tensor_model_parallel_size` | 1 | `gpu_memory_utilization` | 0.5 |
| `temperature` | 1.0 | `fsdp_config.param_offload` | True |
| `rollout.name` | vllm | | |

## A.2   CHAT TEMPLATES

Here we provide chat templates used in our implementation. A part of our implementation is based on AZR (Zhao et al., 2025), therefore the task generation template is similar to that used in AZR.

Template for difficulty prediction:

> ### Difficulty Prediction Requirements:
> - At the end of your generation, you need to review your code and provide a difficulty prediction for the code.
> - The difficulty prediction should be a number between 0 and 8, where 0 is the easiest and 8 is the hardest.
> - The review of your code should be in the <review>...</review> tags. The review should focus on analyzing the difficulty of the code based on the complexity of the code, the number of steps required to solve the problem, the creativity of the code, as well as how powerful the current solver is.
> - You need to control the difficulty of the generated tasks to be medium. A difficulty level at 4 or 5 would be good.
> - The difficulty prediction should be wrapped in <difficulty_prediction> and </difficulty_prediction> tags. It should be strictly a number between 0 and 8. Otherwise, you will be penalized.
> - Therefore, your response should be formatted like
> <think>...</think>   <answer>...</answer>   <review>...</review>
> <difficulty_prediction>...</difficulty_prediction>

Template for task generation:

> ### Task: Create a Python Code Snippet (where custom classes are allowed, which should be defined at the top of the code snippet) with one Matching Input
> Using the reference code snippets provided below as examples, design a new and unique Python code snippet that demands deep algorithmic reasoning to deduce one possible input from a given output. Your submission should include both a code snippet and test input

pair, where the input will be plugged into the code snippet to produce the output, which that function output be given to a test subject to come up with any input that will produce the same function output. This is meant to be an I.Q. test.

### Code Requirements:
- Name the entry function 'f' (e.g., 'def f(...): ...'), you can have nested definitions inside 'f'
- Ensure the function returns a value
- Include at least one input parameter
- Make the function deterministic
- Make the snippet require state tracking across multiple data transformations, ensuring the task requires long multi step reasoning
- AVOID THE FOLLOWING:
* Random functions or variables
* Date/time operations
* I/O operations (reading files, network requests)
* Printing or logging
* Any external state
- Ensure execution completes within 10 seconds on a modern CPU
- All imports and class definitions should be at the very top of the code snippet
- The snippet should end with a return statement from the main function 'f', anything after will be removed

### Input Requirements:
- Provide exactly one test input for your function
- Format multiple arguments with commas between them
- Remember to add quotes around string arguments

### Formatting:
- Format your code with: '''python
def f(...):
# your code here
return ...
'''
- Format your input with: '''input
arg1, arg2, ...
'''

### Example Format:
'''python
def f(name: str, info: dict):
# code logic here
return result
'''

'''input
'John', 'age': 20, 'city': 'New York'
'''

### Evaluation Criteria:
- Executability, your code should be executable given your input
- Difficulty in predicting the output from your provided input and code snippet. Focus on either algorithmic reasoning or logic complexity. For example, you can define complex data structure classes and operate on them like trees, heaps, stacks, queues, graphs, etc, or use complex control flow, dynamic programming, recursions, divide and conquer, greedy, backtracking, etc
- Creativity, the code needs to be sufficiently different from the provided reference snippets
- Restricted usage of certain keywords and packages, you are not allowed to use the following words in any form, even in comments: raise

First, carefully devise a clear plan: e.g., identify how your snippet will be challenging, distinct from reference snippets, and creative. Then, write the final code snippet and its inputs.

## B    THE USE OF LARGE LANGUAGE MODELS(LLMS)

During the preparation of this paper, we use LLMs as a writing assistant to polish up some expressions. Its role was strictly limited to improving grammar, clarity, and readability. The LLM was not used for any substantive aspect of this research, such as the generation of core ideas, the development of the methodology, code implementation, data analysis, or the formulation of conclusions.

