# OpenReview forum: "Self-Aware Reinforcement Learning for Improving LLMs with Minimal Data"
_ICLR.cc/2026/Conference — Submitted to ICLR 2026_

### Official Review · Reviewer_BYJc · 2025-10-27

**Soundness:** 2
**Presentation:** 3
**Contribution:** 2
**Rating:** 4
**Confidence:** 4

**Summary:**

This paper introduces self-aware RL, a paradigm for improving large language models through reinforcement learning with reduced data requirements. The approach features two mechanisms: (1) self-aware difficulty prediction, where the model learns to estimate task difficulty relative to its current capabilities, and (2) self-aware limit breaking, where the model occasionally requests external guidance for high-utility unsolvable tasks. The authors train Qwen2.5-Coder-3B on code generation tasks and evaluate across various benchmarks in mathematical reasoning and code generation, reporting improvements on math benchmarks.

**Strengths:**

- Addresses a problem of reducing data dependency in LLM training.
- The paper is well written and easy to follow.

**Weaknesses:**

Please see my detailed questions and concerns below.

**Questions:**

- What prevents reward hacking? The generator is rewarded for accurate difficulty prediction, but couldn't it hack this by generating only tasks of a narrow difficulty range where its predictions are accurate?
- How sensitive is the method to the target difficulty level mentioned in the prompt (lines 633)? You explicitly tell the model to aim for difficulty 4-5, isn't this just supervised prompting rather than "self-awareness"?
- Why use z-score normalization across the buffer? This makes the selection relative to recent tasks, but what if the buffer contains mostly easy or mostly hard tasks?
- Why measure novelty using perplexity from the old policy $\pi_{\theta_{old}}$ rather than the current policy? Shouldn't novelty be relative to what the current model knows?
- How do you ensure the generated tasks are valid, well-formed, and non-trivial? What quality control mechanisms exist beyond format checking?
- How do you handle the distributional shift as the model improves? Early-generated tasks may become trivial. Do you discard old data or retrain on everything?

---

> ### Author Response · Authors · 2025-11-24
>
> Q.1
>
> We thank the reviewer for the question regarding how task difficulty is controlled.
> As described in Appendix A.2, the generator is explicitly instructed to generate tasks at a suitable difficulty level.
>
> Q.2
>
> We appreciate the reviewer’s interest in the definition of the difficulty metric.
> The difficulty metric is defined to reflect the solver’s accuracy. It is not an absolute measure of task difficulty, but rather a relative measure indicating how difficult a problem is from the perspective of the solver policy. This relative nature is what makes the difficulty prediction “self-aware.” The difficulty level should be controlled around 0.5, which has been validated by previous research works [1, 2].
>
> Q.3
>
> We thank the reviewer for seeking clarification on the design of the difficulty utility.
> The design ensures that the difficulty utility is measured based on the current status of the policy. Comparing a new task to another task proposed 100 rounds earlier would not be reasonable, since the policy may have changed significantly after 100 rounds. The difficulty measurement obtained at that earlier time is no longer valid when the new task is generated.
>
> Q.4
>
> We appreciate the reviewer’s question about the role of $\pi_{\theta_{old}}$.
> $\pi_{\theta_{old}}$ refers to the policy at the start of each batch. Commonly, LLM RL is implemented in a slightly off-policy manner. For example, if the batch_size is 64 and the micro_batch_size is 16, the policy will be updated 64/16 = 4 times on each 1/4 part of the training batch, which produces $\theta_0$ (not updated), $\theta_1$ (updated once), $\theta_2$ (updated twice), $\theta_3$ (updated 3 times), and $\theta_4$ (updated 4 times). $\pi_{\theta_{old}}$ here refers to $\theta_0$. $\pi_{\theta_{old}}$ is close to the actual current model at data collection time, and calculating perplexity based on this model is also easier due to some implementation details.
>
> Q.5
>
> We thank the reviewer for asking about task validity.
> All tasks are validated via a Python code executor. As detailed in Appendix A.1, in SARL we utilize the QWQ executor for this purpose.
>
> Q.6
>
> We appreciate the reviewer’s interest in the data usage protocol.
> Each generated task is used to train the policy exactly once. We do not reuse previously generated tasks.
>
> [1] Chen, X., Lu, J., Kim, M., Zhang, D., Tang, J., Piché, A., ... & Kamalloo, E. (2025). Self-Evolving Curriculum for LLM Reasoning. *arXiv preprint arXiv:2505.14970*.
> [2] Bae, S., Hong, J., Lee, M. Y., Kim, H., Nam, J., & Kwak, D. (2025). Online difficulty filtering for reasoning oriented reinforcement learning. *arXiv preprint arXiv:2504.03380*.

---

### Official Review · Reviewer_4wUA · 2025-10-29

**Soundness:** 2
**Presentation:** 2
**Contribution:** 2
**Rating:** 2
**Confidence:** 3

**Summary:**

The paper introduces a self-aware reinforcement learning framework that allows an LLM to generate and solve its own tasks while using two mechanisms: 1) self-aware difficulty prediction, where the generator predicts task difficulty relative to the solver's ability; 2) self-aware limit breaking, where the model detects when it has hit the capability ceiling and requests external data for targeted improvement. Based on these two mechanisms, the paper designs reward functions and implement on REINFORCE++ algorithm. The paper evaluates the proposed framework with Qwen-Coder-3B model across 9 reasoning benchmarks.

**Strengths:**

1. The paper is well-motivated and addresses an important problem of reinforcement learning for LLMs.
2. The designs for difficulty prediction and adaptive external querying are reasonable and conceptually sound.

**Weaknesses:**

1. It is unclear from Equation (10) how the solver and generator are updated; the equation should explicitly write out the terms for both the solver and the generator to clarify their respective objectives.
2. The reward design for the generator seems too simple. Will the generator produce overly easy or overly difficult questions to maximize the difficulty prediction reward?
3. The experiments are confined to a single model and single run. The reported improvements on code benchmarks, despite training on code data, are marginal. It is hard to determine whether the method is consistently or statistically effective.
4. Lack of justification of choice of $\gamma$.
5. The paper writing should be improved, many typo and format errors are presenting, such as at line 273 and line 477.

**Questions:**

1. What is the rationale for multiplying $R_{format}$ in equation (8) and equation (9)?
2. In Figure 2, why does the training reward become negative, given that the defined rewards are non-negative? Additionally, it would be helpful to show separate curves for the format reward, outcome reward and difficulty prediction reward.

---

> ### Author Response · Authors · 2025-11-24
>
> W.1
>
> We thank the reviewer for the question regarding the RL objective.
> Equation 10 introduces the RL training objective based on the reward design presented earlier in the paper. The RL objective itself is introduced in Section 3. In the revised version, we will add more details and clarification around Equation 10 to make this connection clearer.
>
> W.2
>
> We appreciate the reviewer’s interest in the effectiveness of the reward design.
> The difficulty prediction (DP) reward and rollout accuracy are reported in Figures 3 and 4. Without SARL, the generator initially produces overly simple tasks and thus obtains a low DP reward. However, after being trained for around 50 rounds, we observe clear improvements in the DP reward, and the rollout accuracy becomes reasonable for the solver.
>
> Since this relatively simple generator reward already successfully improves the generator’s ability to generate suitable tasks and predict difficulty, we believe the current reward design is sufficient to achieve our goal.
>
> W.3
>
> We thank the reviewer for suggesting a more detailed quantitative analysis.
> We will report statistical results in the updated version.
>
> The marginal improvement on coding tasks is explained in Lines 353–357. Qwen2.5-Coder-3B is already a strong model for coding, leaving limited room for further improvement. This observation is consistent with previous works [1].
>
> W.4
>
> We appreciate the reviewer’s request for clarification. The discount factor $\gamma$ is fixed as 5. We will explicitly clarify this choice in the updated version.
>
> W.5
>
> We thank the reviewer for carefully checking the manuscript and pointing out these issues.
> We will fix these issues in the updated version.
>
> Q.1
>
> We thank the reviewer for the question regarding the reward formulation.
> This is a simplified way of delivering the reward design. If the response is not correctly formatted (format reward = 0), the policy receives zero reward even if it would otherwise obtain a positive reward from problem solving. When the response is well formatted (format reward = 1), the policy is normally rewarded based on the problem-solving performance.
>
> Q.2
>
> We appreciate the reviewer’s attention to how the reward is reported.
> The reward is shifted to provide negative supervision. We will clarify this point in the updated version.
>
> [1] Zhao, Andrew, et al. "Absolute zero: Reinforced self-play reasoning with zero data." *arXiv preprint arXiv:2505.03335* (2025).

---

### Official Review · Reviewer_cSb4 · 2025-11-04

**Soundness:** 2
**Presentation:** 3
**Contribution:** 2
**Rating:** 6
**Confidence:** 4

**Summary:**

The paper introduces a novel paradigm called Self-Aware RL, which allows the LLM to learn and improve with minimal external data by creating a self-task generation-solver design. To make sure the self-generated tasks are valid, this paper proposes two strategies to strengthen the usefulness of the generated samples. First, Self-Aware Difficulty Prediction: The LLM not only generates its own training tasks but also predicts the difficulty of that task relative to its current ability. It then learns to align this difficulty estimate with its actual success rate, enabling an adaptive curriculum that prioritizes tasks that are appropriately challenging, yet solvable ("just right"). Second, Self-Aware Limit Breaking: The model identifies its own capability limits and proactively seeks external guidance only when necessary to surpass its inherent capability limits.

**Strengths:**

1. This paper counts as a pioneer work on "zero-data" self-improving language models, directly comparing to Absolute Zero and showing decent performance improvement.

2. They validate their approach on a good amount of downstream tasks spanning from math to coding, showing steady improvement and good generalization.

3. They have a good cognitive science motivation behind by explicitly connecting LLM self-improvement to the concept of self-awareness provides a novel theoretical framing for future research into self-evolving agents.

**Weaknesses:**

- The entire system's success depends on the LLM first being able to accurately gauge its own difficulty, and we all know LLMs tend to overestimate their confidence for task difficulties. A poor start in self-awareness could easily cause the RL process to settle on an unproductive local minimum. There is currently no mechanism to break this local minimum loop.

- For the second component design, Self-Aware Limit Breaking, external knowledge guidance is used, which is basically a larger-sized LLM. However, this paper completely ignores a straightforward baseline where directly distilling a very small amount of data from this teacher model could yield competitive performance gain.

- The paper is missing experimental comparison with some closely related literature.

Wang et al. Co-Evolving LLM Coder and Unit Tester via Reinforcement Learning.Zhou
Zhou et all. Self-Challenging Language Model Agent


- Some typos and grammar should be better polished (for example, line 223 "self-aware self-aware RL") and references need to be made more professional.

**Questions:**

Please see weakness.

---

> ### Author Response · Authors · 2025-11-24
>
> W.1
>
> We thank the reviewer for highlighting this aspect of our method.
> SARL is specifically proposed to address this problem. As shown in Figures 3 and 4, the generator initially produces overly simple tasks and therefore obtains a low difficulty prediction (DP) reward, which corresponds to “a poor start in self-awareness.” As training proceeds, SARL trains the generator to become better at estimating its own capability, which is reflected by a higher DP reward (Figure 3) and a more suitable rollout accuracy (Figure 4).
>
> W.2
>
> We appreciate the reviewer’s suggestion regarding distillation.
> Distillation is certainly a promising direction and can yield strong improvements. However, the key of Self-Aware Limit Breaking lies in knowing when to learn from the stronger model. As described in Lines 201–204, it is not efficient to seek external guidance blindly on every unsolvable task.
>
> We will add new results comparing distillation and SARL in the updated version to further clarify their relationship and relative benefits.
>
> W.3
>
> We thank the reviewer for pointing out related work.
> SCA is designed for training LLMs on multi-turn agentic tasks, which departs from the setting considered in SARL. Nevertheless, we agree that a comparison to related curriculum-style methods is valuable, and we will add a comparison to CURE in the updated version.
>
> W.4
>
> We appreciate the reviewer for carefully checking the manuscript and pointing out these issues.
> We will fix these issues in the updated version.

---

### Official Review · Reviewer_pJGc · 2025-11-08

**Soundness:** 2
**Presentation:** 3
**Contribution:** 2
**Rating:** 2
**Confidence:** 4

**Summary:**

This paper proposes a RL framework to improve LLM reasoning with minimal data. In this self-improving framework for LLM reasoning where generator agent generates the tasks and predicts success rate and solver agent tries to solve the task and selectively queries a stronger external solver when valuable unsolved tasks are identified.
Two mechanisms are introduced:
1. self-aware difficulty prediction to align task generation with model capability
2. self-aware limit breaking where agent request external guidance for high-utility failures tasks.

**Strengths:**

1. Practical and relevant direction toward data-efficient RL for reasoning models
2. Strong empirical improvements on multiple math reasoning datasets. Evaluated on wide number of benchmarks.

**Weaknesses:**

1. “self-aware” term is conceptually overstated. success-rate prediction alone does not fully justify the terminology.  the framework does not model its internal reasoning or belief states. A clearer positioning as capability-aware curriculum RL would improve conceptual accuracy.
2. In limit-breaking claim, this paper does not introduce a new learning paradigm or theoretical formulation, it simply adds gating logic to decide when to query an external model. Any performance gains depend on the external model’s capability.
3. Evaluation only compares against the base model and AZR, omitting other strong curriculum RL baselines.
4. this framework only tested on qwen-3B model, why not other model families? Have the authors tested different scales or model families? How would the approach transfer to other model families (e.g., Llama, DeepSeek)?

**Questions:**

Please refer to weaknesses

---

> ### Author Response · Authors · 2025-11-24
>
> W.1
>
> We thank the reviewer for pointing out the connection to curriculum RL.
>
> SARL is indeed related to curriculum RL. However, it is not a standard form of curriculum RL, since SARL does not handle a single fixed set of tasks under similar limitations or resources. Instead, SARL operates over dynamically generated tasks with different characteristics and constraints.
>
> W.2
>
> Thank you for raising this point. We agree that our work does not introduce an entirely new learning paradigm or a new theoretical framework. Our goal is not to propose a new learning paradigm, but rather to demonstrate that carefully learned routing between a base model and an external model can substantially surpass the conventional performance upper bound of the base model. The key contribution lies in learning *when* to query the external model, thereby trading off training efficiency against performance gains.
>
> W.3
>
> We appreciate the reviewer’s continued interest in the relation to curriculum RL.
>
> As explained in the response to W.1, SARL is aligned with the general spirit of curriculum RL but differs in important ways, particularly in how tasks and resources are handled. For this reason, SARL is not directly comparable to standard formulations of curriculum RL.
>
> W.4
>
> We thank the reviewer for the suggestion to include more results on different base models.
>
> We are adding additional results on Llama in the revised version. For DeepSeek, however, it is very challenging to tune 671B-parameter models with only 4 GPUs (our hardware configuration is provided in Appendix A.1 for reference). Therefore, we will not consider adding more results on the DeepSeek series at this stage.

---

### Official Review · Reviewer_jm8t · 2025-11-12

**Soundness:** 2
**Presentation:** 3
**Contribution:** 2
**Rating:** 4
**Confidence:** 3

**Summary:**

This paper addresses data‐efficiency in RL‐fine­tuning of large language models (LLMs). The authors propose a “self‑aware RL” framework in which an LLM alternately (i) generates tasks and predicts their difficulty (self‑aware difficulty prediction), and (ii) attempts to solve them, with a mechanism to detect when tasks are beyond current capabilities and ask for external guidance (self‑aware limit breaking). The idea is to minimize reliance on large curated datasets by letting the model generate its own curriculum and seek external data only when necessary.

**Strengths:**

* The problem is timely: reducing dependence on curated data for reasoning-capable LLM fine-tuning is important.

* The idea of alternating roles (task generation / solving) is conceptually appealing and aligns with recent trends in self-supervision and synthetic data generation.

* The empirical results show non-trivial gains with relatively little extra data, which is promising for data-efficiency.

**Weaknesses:**

* The proposed framework is essentially a form of adversarial or dual-role training (generator and solver), yet the paper lacks analytical treatment of convergence, stability, or whether the iterative procedure reliably improves rather than degenerate (e.g., the generator produces trivial tasks, the solver overfits, task difficulty drifts). There is no deeper theoretical justification or monitoring of regime stability.
* My own experience (and known community experience) is that even SOTA LLMs generating tasks often yield unstable data (invalid tasks, unsolvable ones, or trivial ones). The paper does not sufficiently discuss how data quality is controlled, how “ground truth” solutions are validated (especially when the same model solves its own generated tasks), or how noisy/invalid samples are filtered. This raises questions about how much of the reported improvement is due to good synthetic data vs incidental or artifact effects.
* The “multiple model roles” (generator + solver + possibly verifier) increases complexity. The paper does not sufficiently discuss the scaling behaviour (model sizes, compute budget, how many rounds, etc.). It is unclear whether this method is practical for much more parameter models.

**Questions:**

* Could you include training curves (e.g., generator loss, solver loss, number of generated tasks, accuracy of solver over time) to illustrate training dynamics and stability?

* Is the method truly free of external data (i.e., “no external data” apart from initial model and minimal seeds)? If so, how is grounding / calibration ensured? If not, how much external data is needed and what is its role?

* Please report compute / training cost: number of model parameters used, number of rounds of generator/solver, GPU hours, wall-clock time. What is the overhead compared to standard fine-tuning?

---

> ### Author Response · Authors · 2025-11-24
>
> W.1
>
> We thank the reviewer for raising this concern and are happy to clarify our design choice.
> First, SARL is not formulated as an adversarial training paradigm. Furthermore, as shown in Figure 4, we carefully monitored and analyzed the training dynamics of SARL. Specifically, the generator initially tended to generate trivial (overly easy) tasks, and this issue was subsequently corrected by the difficulty prediction component. Figure 4 clearly reflects this trend.
>
> W.2
>
> We appreciate the reviewer’s attention to the validity and solvability of the generated tasks.
> As described in Appendix A.1, all coding tasks are validated through a QWQ executor, which ensures that the generated tasks are valid and solvable.
>
> W.3
>
> We thank the reviewer for the thoughtful comment regarding the complexity of our method.
> The external solver is invoked for only 1% of the queries, which does not substantially increase the overall complexity of the system.
>
> Noteworthily, the data collection procedure in previous RLVR paradigms is far from “not complex”. Considerable effort was required for data collection, synthesis, labeling, filtering, and even iterative retraining based purely on experience. In contrast, the generator in SARL takes over the human’s role by dynamically preparing the training data. The perceived “increased” complexity mainly arises from overlooking the inherently complex data preparation procedures in previous works, which are typically not counted as part of RL training, even though they are essential for making the training feasible.
>
> Q.1
>
> We thank the reviewer for the detailed questions regarding the training process and statistics.
> The accuracy of the solver is reported in Figure 4, and the training reward is reported in Figure 2. The number of generated tasks is 64 questions per round. As detailed in Appendix A.1, we trained the model for 200 steps, which in total incorporated 12,800 tasks.
>
> Q.2
>
> We appreciate the reviewer’s interest in the data sources and calibration process.
> No external data are used apart from the initial model and a minimal set of seed tasks. Calibration is ensured by the environment validation, which guarantees that the solver is trained only on valid and solvable tasks, with reliable reward supervision.
>
> Q.3
>
> We thank the reviewer for asking about the computational setup and comparison baseline.
> We fine-tuned all parameters of Qwen2.5-Coder-3B, which means the “number of model parameters used” is 3 billion. The number of rounds for the generator/solver, as detailed in Appendix A.1, is 200. A standard training pipeline typically takes around 36 hours on 4 NVIDIA H200 GPUs.
>
> Regarding “standard fine-tuning,” we would appreciate clarification on whether this refers to supervised fine-tuning (SFT) or to a commonly used RL paradigm. We believe SARL is not directly comparable to either of these paradigms, as SARL explicitly incorporates external data preparation into the training pipeline, whereas such data preparation is usually treated as a separate, pre-training process in standard setups.

---

### Meta-Review · Area_Chair_o4oF · 2026-01-12

**Summary:**

The paper proposes a self-aware reinforcement learning framework that enables LLMs to generate and solve their own tasks through two mechanisms: self-aware difficulty prediction (where the model predicts task difficulty relative to its capabilities) and self-aware limit breaking (where the model selectively requests external guidance for high-utility unsolvable tasks). Performance gains in RL training were demonstrated.

Reviewers raised many concerns about this paper, including but not limited to: (1) lack of theoretical grounding and stability/convergence analysis, (2) potential reward hacking, (3) scalability to larger models, (4) missing comparisons to distillation and curriculum RL baselines, (5) the conceptual accuracy of the "self-aware" terminology, (6) single-model evaluation with no statistical validation. The AC believes most of the concerns are valid.

**Reviewer Concerns:**

The author addressed some concerns and answered some questions during the rebuttal period, such as training dynamics and curves, computational cost, task validation mechanism via QWQ executor, and hyperparameter choices. Some of these were included in the the appendix. However, several key concerns were not addressed, including (1) No theoretical analysis of convergence, stability, or learning guarantees; (2) Missing baselines; and (3) Statistical validation and stability concerns over larger and a wide range of models.

**Reviewer Scores:**

Based on the rebuttal discussion and outstanding concerns, it is unlikely that the Reviewer pJGc and 4wUA will flip their scores to support this paper, as the key concerns mentioned above remain and the new empirical evidence provided by the authors are limited.

---

### Decision · Program_Chairs · 2026-01-26

Reject